# Experimental transfusion of variant CJD-infected blood reveals previously uncharacterised prion disorder in mice and macaque

Emmanuel E. Comoy[1], Jacqueline Mikol[1], Nina Jaffré[1,2], Vincent Lebon[1], Etienne Levavasseur[3], Nathalie Streichenberger[4], Chryslain Sumian[2], Armand Perret-Liaudet[4], Marc Eloit[5], Olivier Andreoletti[6], Stéphane Haïk[3], Philippe Hantraye[1] & Jean-Philippe Deslys[1]

Exposure of human populations to bovine spongiform encephalopathy through contaminated food has resulted in <250 cases of variant Creutzfeldt–Jakob disease (vCJD). However, more than 99% of vCJD infections could have remained silent suggesting a long-term risk of secondary transmission particularly through blood. Here, we present experimental evidence that transfusion in mice and non-human primates of blood products from symptomatic and non-symptomatic infected donors induces not only vCJD, but also a different class of neurological impairments. These impairments can all be retransmitted to mice with a pathognomonic accumulation of abnormal prion protein, thus expanding the spectrum of known prion diseases. Our findings suggest that the intravenous route promotes propagation of masked prion variants according to different mechanisms involved in peripheral replication.

[1] CEA, Institut François Jacob, Université Paris-Saclay, 18 Route du Panorama, 92265 Fontenay-aux-Roses, France. [2] MacoPharma, 200 Chaussée Fernand Forest, 59200 Tourcoing, France. [3] Université Pierre et Marie Curie, UMR-S 1127, CNRS UMR 722, Institut du Cerveau et de la Moelle Epinière, G.H. Pitié-Salpêtrière, 47 Boulevard de l'Hôpital, 75013 Paris, France. [4] Hospices Civils de Lyon, Université Claude Bernard Lyon 1, Institut NeuroMyogène CNRS UMR 5310—INSERM U1217, 59 Boulevard Pinel, 69677 Bron, France. [5] Institut Pasteur, 15 Rue du Docteur Roux, 75015 Paris, France. [6] UMR INRA-ENVT 1225, Ecole Nationale Vétérinaire de Toulouse, 23 chemin des Capelles, 31076 Toulouse, France. Correspondence and requests for materials should be addressed to E.E.C. (email: emmanuel.comoy@cea.fr)

Bovine spongiform encephalopathy (BSE) was widely distributed in Europe and most notably in the UK. Overall, more than 2 million undiagnosed infected cattle would have entered the human food chain[1], implying an estimated exposure of 10 million consumers[2]. However, the fatal variant Creutzfeldt–Jakob disease (vCJD) in humans has fortunately remained rare, with 177 cases in the UK to date (51 in the rest of the world, including 27 in France); the majority of cases were detected between 1995 and 2005, and only three cases from 2012 to 2016. Besides primary contamination through food, inter-human vCJD transmission has been associated with blood transfusions from donors incubating vCJD in the UK, but this has been limited to 4 cases among the 67 recorded recipients of blood from identified vCJD donors (<6%)[3]. In addition, one possible silent infection was associated with treatment with Factor VIII extracted from a pool of 10,000 plasma donations that included plasma from a preclinical case of vCJD[4]. Such contamination occurred before the introduction of systematic deleukocytation procedures in 2003[5] and improvements in techniques for the purification of plasma derivatives. No transmission of vCJD by transfusion has been reported since the implementation of those measures (blood infectivity is presumed to be shared in a ratio of 50:50 between leucocytes and plasma[6]).

In contrast to the limited prevalence of clinical vCJD, abnormal prion protein (PrP[d]) was detected in 16 appendices among more than 32,000 samples in a recent anonymized survey of prevalence in the UK population. This suggests an unexpectedly high prevalence (1/2000, 493 per million) of asymptomatic infected subjects that is at least 100-fold higher than that of reported clinical cases[7]. The development of the disease after an extended period of time, may be several decades as observed with Kuru[8] and iatrogenic CJD[9], cannot be excluded. However, the most probable scenario is that after exposure to BSE, more than 99% of infected individuals will remain as silent carriers, possibly for decades. Such a long-term silent infection of lymphoid tissues represents a risk of secondary transmission that is difficult to assess. With regard to blood transfusion, basic calculations suggest that the risk may be in the order of 1000 potentially contaminated blood donations every year in the UK. Moreover, projections assume that the prevalence of clinical and subclinical infections in most of the other countries of Western Europe would be significant, since

no more than 10- to 100-fold lower[10]. According to these levels of exposure, which are much higher than those of several blood-borne viruses, there is continuing cause for concern about the management of blood, blood components and derivatives, and surgical instruments that supports further evaluation of risks from primary BSE and secondary vCJD in relevant experimental models.

In this study, we aim to explore the risk of prion infection from blood-derived products using both conventional mice susceptible to vCJD[11,12] and in non-human primates that are considered to be the ultimate model of the human condition with regard to prions[13,14], especially for BSE infection[15,16]. Neurological impairments are transmitted to recipients in these two animal models using blood infectivity (transfusion of blood products derived from infected donors) and brain infectivity (intravenous exposure to soluble infected material derived from brain tissue). Nevertheless, a low percentage of the affected recipient animals exhibit specific features of vCJD, whereas a much higher proportion of them develops neurological impairments devoid of the classical biochemical and/or pathological features that currently constitute the diagnostic criteria for prion diseases. These atypical impairments are identified as prion disorders by secondary transmission.

## Results

**Transmission of neurological impairments without PrP[res]**. In a series of independent experiments, we transfused blood products derived from donors exposed to vCJD into 470 wild-type mice and 19 cynomolgus macaques. In parallel, we injected 70 mice and 11 macaques intravenously (IV) with soluble brain extracts derived from BSE- or vCJD-infected donors; the extracts were obtained according to protocols designed to mimic blood infectivity[17]. As negative controls, animals were sham-inoculated (40 mice and 6 macaques) or exposed to blood or brain products from healthy donors (90 mice and 2 macaques), while positive controls included animals exposed to crude homogenates of infected brains through the intracerebral (IC) (38 mice and 10 macaques) or IV (12 mice and 4 macaques) route.

All the IC- and IV-inoculated positive control animals (50/50 mice and 14/14 macaques) developed vCJD (Fig. 1). Protease-

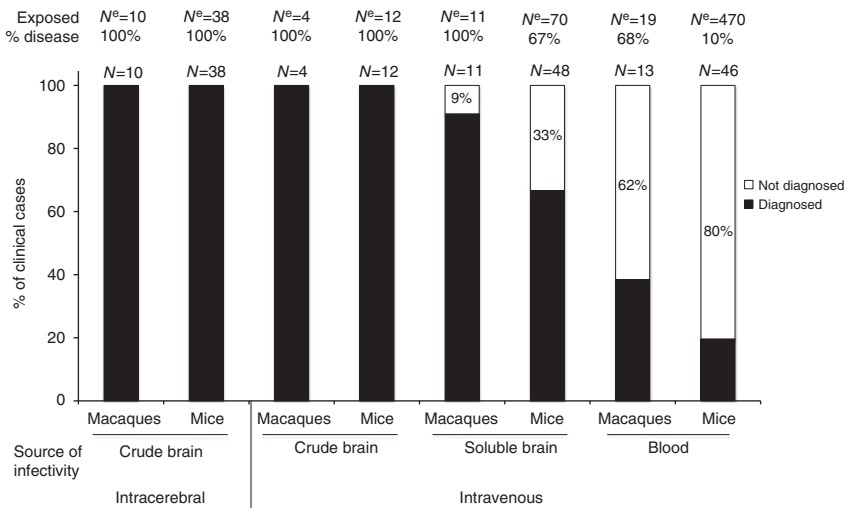

**Fig. 1** Ability of current diagnostic criteria to identify the disorders as prion diseases as observed in the different models used in this study. We considered here that the diagnosis of prion disease is based on the presence of PK-resistant abnormal PrP in the brains of animals with possible prion disease. For each model, $N^e$ corresponds to the number of animals exposed to the sources of infectivity and $N$ to the number of mice developing neurological impairments (% disease = $N/N^e \times 100$). The numbers within bars correspond to the percentages of neurologically affected animals without detectable PrP[res]

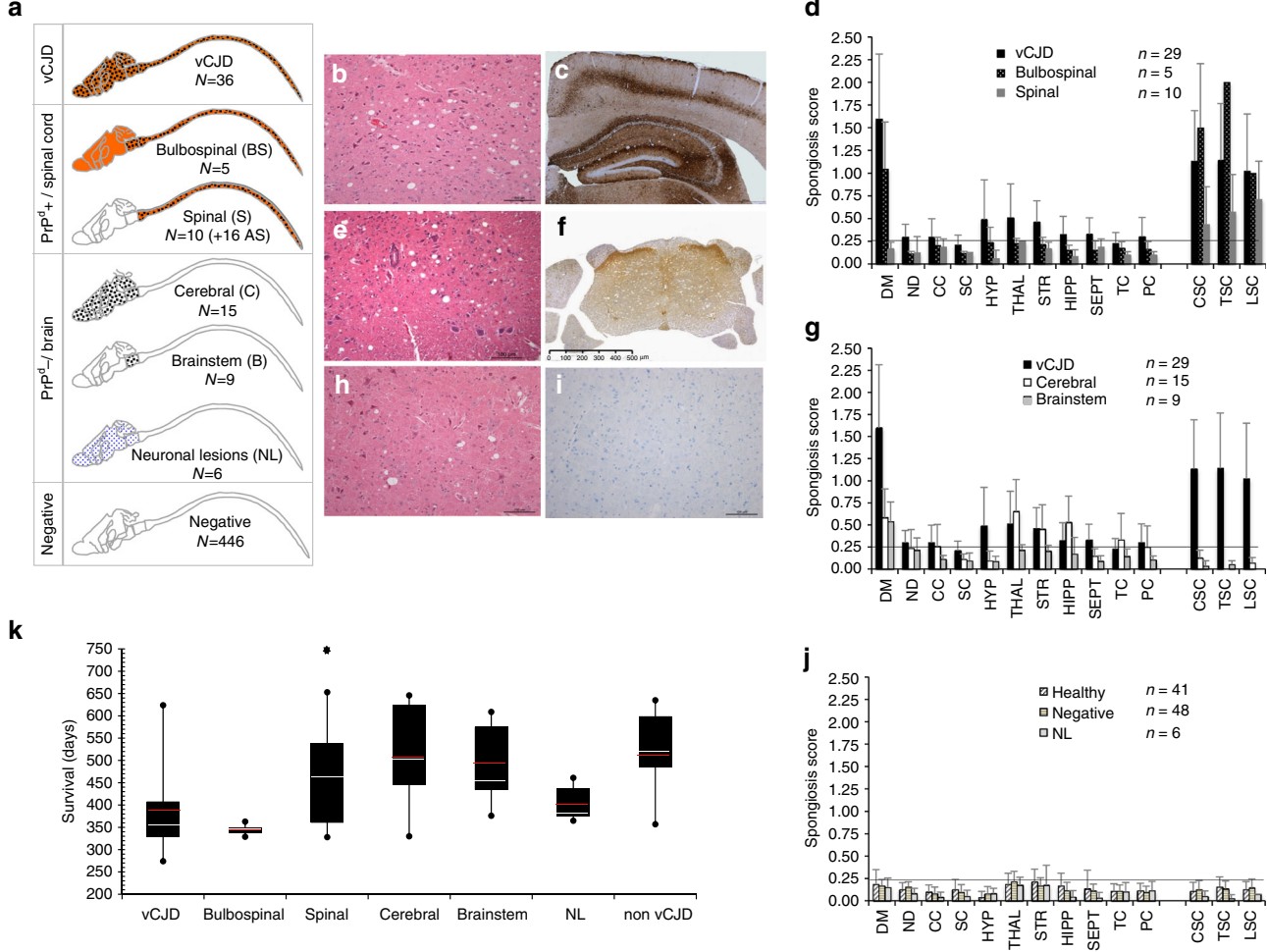

**Fig. 2** Distinct disease phenotypes observed in affected mice following intravenous exposure to vCJD-contaminated products. **a** Synopsis of the distribution of lesions and biochemical features in the central nervous system (CNS) of mice depending on the disease phenotype. Small dots indicate the presence of only neuronal lesions (mainly vacuolation), large dots indicate spongiform change (plus neuronal lesions) and the orange colour indicates the presence of abnormal PrP[d]. The respective numbers of cases are reported (N). vCJD phenotype: **b** classical spongiform change (hemalun–eosine, H&E stain, 200×) and **c** abnormal PrP[d] deposition (Saf-32 anti-PrP antibody, 50×). Bulbospinal phenotype (BS): **d** spongiform changes were limited to the brainstem and spinal cord without cerebral involvement. Spinal phenotype (S): **e** spongiform change was limited to part or all of the spinal cord (H&E stain, 200×) but **d** no obvious brain lesion. **f** Accumulation of PrP[d] in the spinal cord and dorsal roots was detected by immunohistochemistry in those 10 animals and the 16 aging spinal (AS) mice. Cerebral phenotype (C): moderate neuronal lesions and **h** cerebral spongiform change (H&E stain, 200×) with **g** a topographical distribution similar to vCJD mice (among them, 9 animals exhibited spongiform change limited to the brainstem, and were sub-grouped under the brainstem phenotype, or B). **i** They exhibited no accumulation of peripheral or central PrP[d] (Saf-32 anti-PrP antibody, 200×) that was detectable using conventional techniques (ELISA, western blot and/or immunohistochemistry). Neuronal lesions only phenotype (NL): six animals showed apoptotic neurons with vacuolation in the brain (**j**) without spongiform change or detectable accumulation of PrP[d]. Mean (+standard deviation) spongiosis profiles (**d**, **g**, **j**) observed for the different disease profiles. The horizontal line corresponds to the limit of significance. DM dorsal medulla, ND nucleus dentatus, CC cerebellar cortex, SC superior colliculus, HYP hypothalamus, THAL thalamus, STR striatum, HIPP hippocampus, SEPT septum, TC temporal cortex, PC parietal cortex, CSC cervical spinal cord, TSC thoracic spinal cord, LSC lumbar spinal cord. **k** Survival time distribution of neurologically affected mice according to their disease phenotypes. Red and white lines indicate the means and medians, respectively (the black star over spinal phenotype correspond to the 16 AS animals)

resistant PrP[d] (PrP[res]) was detected in their brains using classical biochemical diagnostic methods (ELISA and western blot), as a confirmation of their positive prion infection status, as expected. None of the negative control animals exhibited neurological signs.

After intravenous exposure to exogenous preparations (soluble infected brain extracts) or endogenous preparations (infected blood), a proportion of the animals (94/540 mice and 24/30 macaques) developed fatal neurological impairments with incidences ranging from 10 to 100% depending upon the inoculum. A higher incidence of fatal neurological impairment was obtained in animals exposed to soluble infected brain extracts that were expected to harbour more infectivity than the infected

blood. Among the affected animals, classical biochemical diagnostic methods detected PrP[res] in the brains and spleens of only 41/94 mice and 15/24 macaques (and in none of the asymptomatic animals). We further investigated the neurologically affected animals but PrP[res] negative (53/94 mice and 9/24 macaques) that corresponded, (depending on the inoculum), to between 9 and 80% of the animals developing neurological impairments (Fig. 1).

**vCJD prions induce distinct incomplete vCJD profiles in mice.**
The 94 neurologically affected mice exposed to infected brain

**Table 1 Comparison of neurological signs, pathological and biochemical features of the BSE/vCJD vs. myelopathic syndrome in primates**

| | BSE/vCJD | Myelopathy |
|---|---|---|
| *Neurological signs* | | |
| Behaviour | Aggressiveness | No obvious modification |
| Tremors | Important permanent, increased during movement | Inconstant, very subtle |
| Ataxia | Cerebellar ataxia<br>Loss of equilibrium | Ataxia of limbs |
| Sensory symptoms | Hyperreactivity (jump without habituation following visual or auditory stimulus) | No evidence for hyperreactivity<br>Impaired precise vision |
| Sensitive symptoms | Apparent exacerbated hyperaesthesia of limbs | Apparent exacerbated hyperaesthesia of limbs |
| Motricity | Uncoordinated locomotion | Dysmetria/fine motor impairment of upper limbs (animals failed to catch tiny dry grapes but not balls, then systematically caught food with their mouths), followed by progressive paresis (vacuum cleaner feeding) |
| *Lesions* | | |
| Hemispheres | Spongiform change, gliosis, neuronal lesions | No obvious lesion<br>Wallerian degeneration of optic tracts |
| Cerebellum | | Granule cells moderately rarefied |
| Medulla oblongata | | Bilateral necrotic lesions of spinal nuclei of trigeminal nerve |
| Spinal cord | | Bilateral necrotic lesions of anterior horns (lower cervical cord)<br>Wallerian degeneration of gracilis funiculi |
| *PrP* | | |
| Brain | PK-resistant PrP$^d$ detectable with all the techniques | No detectable PrP$^d$ |
| Spinal cord | PK-resistant PrP$^d$ detectable with all the techniques (grey matter) | No detectable PrP$^d$ (except R5 primate) |

extracts or infected blood exhibited similar neurological signs (general weakness, paresis to paralysis of the lower limbs and tail) to the 50 positive control animals. This was independent of the inocula they received and their expression of PrP$^{res}$; these animals were thus initially suspected of developing the same prion disease as the positive controls. Seventy-four (80%) of the mice with neurological impairments and similar numbers of mice exposed to infected material and devoid of neurological signs (64 mice considered as apparently negative) and control mice (68 mice including 41 negative control mice considered as healthy) were sampled for histology and for detection of PrP$^d$ by immunohistochemistry (Supplementary Table 1).

Among the 94 neurologically affected mice, 29 mice (all PrP$^{res}$ positive) presented a classical vCJD profile: they exhibited significant neuronal lesions (mainly neuronal vacuolation), cerebral and spinal spongiform change (Fig. 2a, b) associated with the accumulation of abnormal PrP (PrP$^d$) in both brain and spinal cord (Fig. 2c) and in their spleen (Supplementary Fig. 1). Seven other animals, that showed accumulation of PrP$^{res}$ but were not sampled for histology, were classified as vCJD (total number of vCJD cases was thus 36).

The 58 other mice exhibited atypical disease phenotypes distinct from vCJD after similar or even longer incubation periods (Fig. 2k). Those disorders appeared as truncated vCJD phenotypes as they presented similar lesions but not the complete spectrum. We classified the animals with atypical neurological impairments into five distinct pathological phenotypes (Fig. 2a) according to their main histological lesions, which may be themselves divided into two main groups according to the presence or absence of PrP$^d$.

The first group included the five remaining PrP$^{res}$ positive animals that we classified as bulbospinal (BS): they exhibited the same pattern of neuronal lesions and PrP$^d$ deposits in the whole central nervous system (CNS) as vCJD mice, but spongiform change was restricted to the brainstem and spinal cord (Fig. 2a, d). An accumulation of PrP$^d$ in their spleen was detected by biochemical and immunohistochemical methods

(Supplementary Fig. 1). In parallel, 10 PrP$^{res}$ negative animals exhibited deposition of PK-sensitive PrP$^d$ (accumulation was observed with immunohistochemistry but biochemistry was negative) and spongiform change both of which were strictly limited to their spinal cords (Fig. 2e, f). These animals were thus classified as spinal (S). It was puzzling to observe that 16 supplementary animals, all exposed to blood products from non-human primates, were still alive and not showing neurological impairments at the end of the study (>750 days), but also exhibited accumulation of spinal PrP$^d$, and among them seven had associated spongiform change. We did not, however, include these animals in the total of S phenotypes according to their ages, but classified them as aging spinal, or AS, animals. Those 15 PrP$^d$ positive (+16 AS) animals constitute the first group of atypical phenotypes with lesions focused on the spinal cord.

The second group comprised 30 animals that showed lesions restricted to the brain (no lesions of the spinal cord) and an absence of detectable PrP$^d$ (Fig. 2g–j). Fifteen of this group exhibited a lesion profile identical to vCJD mice (cerebral phenotype, C), nine animals showed spongiform change limited to the brainstem (brainstem phenotype, B) and six animals only exhibited neuronal vacuolation (neuronal lesions only phenotype, NL).

The 13 remaining PrP$^{res}$ negative animals with neurological signs could not be sampled for histology and thus cannot be specifically subclassified within the S, C, B or NL phenotypes. They were thus classified as non-vCJD. No significant neuronal lesions or spongiform change were observed either in 41 healthy negative control mice sampled for histology (Fig. 2j), sampled at ages corresponding to 171–795 days of incubation, or in 48 among the 446 inoculated animals devoid of neurological signs and considered to be negative (214–762 dpi).

**Unexpected syndromes in macaques after transfusion.** In cynomolgus macaques (see Supplementary Note 1 and Supplementary Fig. 2 for experimental details), the 29 PrP$^{res}$ positive animals exhibited the expected BSE/vCJD phenotype (Table 1;

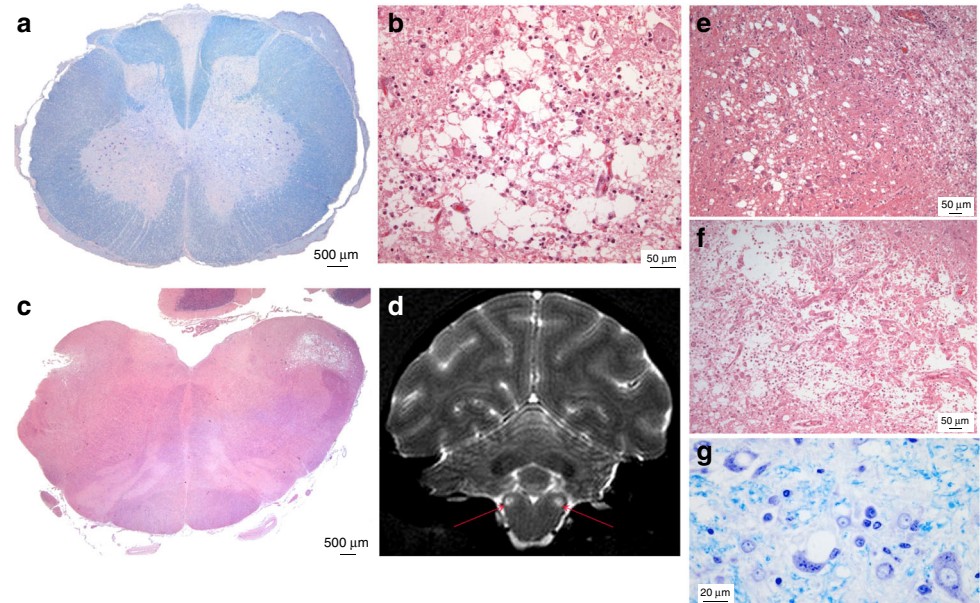

**Fig. 3** CNS lesions in atypical primate infections. **a** Bilateral, medial pseudo-necrotic lesions of the anterior horns with neuronal loss in the lower cervical spinal cord extended in some cases up to C4 (Primate R8. C6 spinal cord. Klüver–Barrera/KB stain). Wallerian degeneration of the posterior spinal tracts (mainly gracilis) was also observed along the whole length of the spinal cord and in parts of the optic tracts (Supplementary Fig. 4). Atrophic ganglion cells were noted in dorsal root ganglia. **b** At higher magnification, lesions appeared as a loss of ground substance filled with macrophages and surrounded by glial cells with few preserved neurons (haematoxylin–eosin/H&E stain). There were no infiltrates of lymphocytes or granulocytes, haemorrhages, fibrin deposits, vascular obstructions or thickened vessels. **c** The same type of bilateral pseudo-necrotic lesion was observed in the brainstem within the spinal nuclei of the trigeminal nerve and extended in some cases to the inferior cerebellar peduncles (Primate R6, H&E stain). **d** Those lesions (arrows) might be visualised with MRI analysis (primate R16, terminal stage of the disease). Coronal T2-weighted images were acquired using a fast spin-echo sequence (TE/ TR = 60/4000, 450 × 450 μm in plane resolution, 1 mm slice thickness, 40 slices, 14 min acquisition time). In these foci, involvement of the nervous tissue was of different degrees, ranging from status spongiosus like (**e**) (primate R16, H&E) to cavitation with myelin debris and axonal loss (**f**) (primate R6, H&E). **g** At the periphery of the lesions, intracellular vacuoles were detected in some neurons (primate R17, KB stain)

Supplementary Fig. 3). They included 24 recipients of brain homogenates and 5 of the 19 transfused macaques (26%) that were all exposed to non-deleukocyted blood products from primate donors with high levels of peripheral PrP[d]. These results confirmed the transmissibility of vCJD by transfusion in primates.

In contrast to the mice, the 9 PrP[res] negative macaques developed a distinct pattern corresponding to an, as yet to our knowledge, undescribed fatal neurological impairment devoid of the main classical features of vCJD. This disorder has the features of a spinal cord disease that is not usually linked to prion disease. The macaques were exposed either to soluble infected brain material (1/11 animals (9%)) or were transfused (8/19 = 42%) with blood derived from donors with detectable accumulation of PrP[d] or infectivity in their peripheral organs, including donors with the status of healthy carriers (Supplementary Note 1). Four other animals exposed to some of those blood products still remain asymptomatic 15.5–17 years post inoculation. No neurological impairment was recorded in the four macaques exposed either to blood from healthy donors (negative controls) or to blood samples with no expected infectivity (derived from primate donors infected intracerebrally with BSE/vCJD and devoid of detectable peripheral PrP[d]).

The 9 macaques that developed this atypical neurological impairment began their long clinical phases (2–6 months) with impaired visual acuity, dysmetry of the forelimbs and they gradually stopped using their hands (Table 1). Hind-limb ataxia appeared after several weeks, followed by progressive proximal paresis of the forelimbs and atrophy of shoulder muscles. Cranial nerve involvement was suspected because of a permanently open mouth, frequent yawning and impaired mobility of the tongue.

The animals exhibited almost no lesions in the cerebral hemispheres: the cortices and basal ganglia were virtually spared; there was moderate rarefaction of cerebellar granule cells but no spongiform change was observed. However, all animals presented severe bulbar and spinal lesions (Fig. 3; Supplementary Fig. 4) and we classified this original pattern as myelopathy.

None of the classical immunohistochemical or biochemical methods, including the latest amplifying methods that are able to detect ultra-low levels of aggregated abnormal PrP with seeding activity (PMCA and RT-QuiC), was able to detect PrP[d] in the CNS (brain or spinal cord) of any of the nine myelopathic macaques (Fig. 4; Supplementary Figs. 5, 6) except in the macaque that was exposed to the highest blood infectivity (macaque R5 in Figs. 4, 5). At the same time, all tests designed to detect an alternative aetiology to a prion disease for this myelopathy were negative (Supplementary Note 2; Supplementary Tables 2–4).

**Primate myelopathy induces incomplete vCJD profiles in mice.** Blood samples (plasma or buffy coat, BC) from PrP[d] negative and PrP[d] positive myelopathic macaques were injected IV into 72 and 71 Swiss mice, respectively, while similar blood products derived from vCJD macaques and mice were injected into 222 and 105 Swiss mice, respectively. Altogether these mice correspond to the 470 mice exposed to blood products as described above. In parallel, mice were inoculated by intracerebral injection with brain material derived from myelopathic and vCJD macaques (N = 10 and 12, respectively).

Neurological impairment was observed in some mice exposed to brain (5/10, 50%) or blood (8/143, 6%) samples derived from the myelopathic macaques: none of the mice developed a

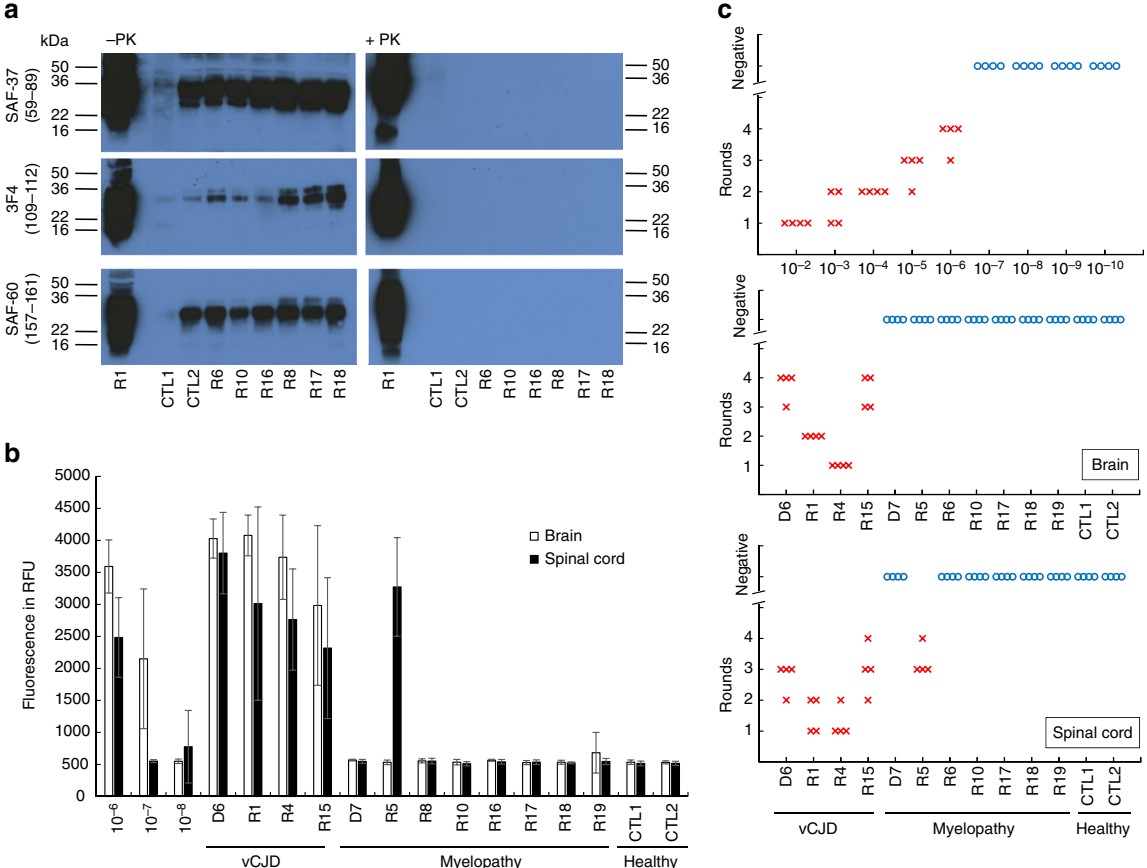

**Fig. 4** Biochemical detection of abnormal PrP in CNS samples of primates. **a** Direct detection of PrP accumulation in spinal cord samples from primates after purification in the absence (−PK) or in the presence (+PK) of proteolysis (40 µg/ml proteinase K for 10 min) detected with 3F4, Saf-37 and Saf-60 antibodies. The equivalent of the same amount of material (8 mg) was deposited in each line. Corresponding uncropped western blots are shown in Supplementary Fig. 5. **b** Amplification of abnormal PrP by RT-QuiC in brain (white bars) or spinal cord (black bar) primate samples. The mean fluorescence from six replicates (two independent experiments with triplicates) after 50 h of amplification is presented here. **c** Amplification of abnormal PrP by PMCA reaction in brain or spinal cord primate samples. Four individual replicates of each sample, including serial dilutions of positive control, were tested. For each sample at each round, the PrP$^{res}$ positive replicates are figured (red crosses). The blue circles figure the negative replicates after five runs. D6, R1, R4 and R15: vCJD primates; D7, R5, R6, R8, R10, R16, R17, R18, R19: myelopathic primates; CTL1, CTL2: healthy primates. Details on each primate are mentioned in Supplementary Note 1

complete vCJD profile, but they exhibited the same incomplete vCJD phenotypes as those described above, including the spinal (S) phenotype with PrP$^d$ accumulation in their spinal cords (Figs. 6, 7, details in Supplementary Table 5). These groups also included 62% (10/16) of the AS animals.

Among the four groups of mice exposed to plasma from the different sources, no statistically significant difference ($\chi^2$-test) was observed in terms of rates of transmission, of PrP$^d$ positive or PrP$^{res}$ positive animals, suggesting that infectivity from plasma is similar in vCJD and myelopathic macaque donors. Conversely, a gradient for the same criteria was observed in the groups exposed to BC samples derived from vCJD mice >vCJD macaques >PrP$^d$ positive myelopathic animals (only three AS mice) >PrP$^d$ negative myelopathic animals, for which no disease was transmitted: buffy coats from myelopathic macaques appeared to harbour limited infectivity, if any.

**Incomplete vCJD profiles are transmissible prion diseases.** Transmissibility of incomplete vCJD phenotypes was assessed through IC or IV injection of material from brain, spinal cord and spleen derived from some mice (Fig. 7b; Supplementary Table 5) within the same mouse strains. Almost complete transmission was obtained with the PrP$^{res}$ positive phenotypes (vCJD and BS),

with recipients exhibiting PrP$^{res}$ after short incubation periods (Supplementary Fig. 7).

PrP$^{res}$ negative phenotypes also transmitted neurological impairments but with lower rates: the diseased animals exhibited the same incomplete phenotypes but also vCJD, confirming that all the incomplete phenotypes are true prion diseases (see details in Supplementary Note 3). Notably the PrP$^d$ positive S phenotype was transmissible and maintained by intracerebral inoculation of CNS tissue or by intravenous administration of spleen tissue.

Among all the transmission studies, the intracerebral route resulted in the preferential development of the complete vCJD phenotype, whereas the intravenous route also induced vCJD but favoured the maintenance of the incomplete phenotypes and notably preferential involvement of the spinal cord (Supplementary Table 6; Supplementary Fig. 8).

**Soluble infectivity promotes PrP$^{res}$ negative prion diseases.** We combined information on the prion nature of the different neurological impairments, and then detailed the distribution of the different phenotypes within the animals exposed to brain or blood infectivity. The results of this analysis clearly showed that the emergence of atypical prion disease phenotypes correlated with the exposure to soluble infectivity derived from brain or

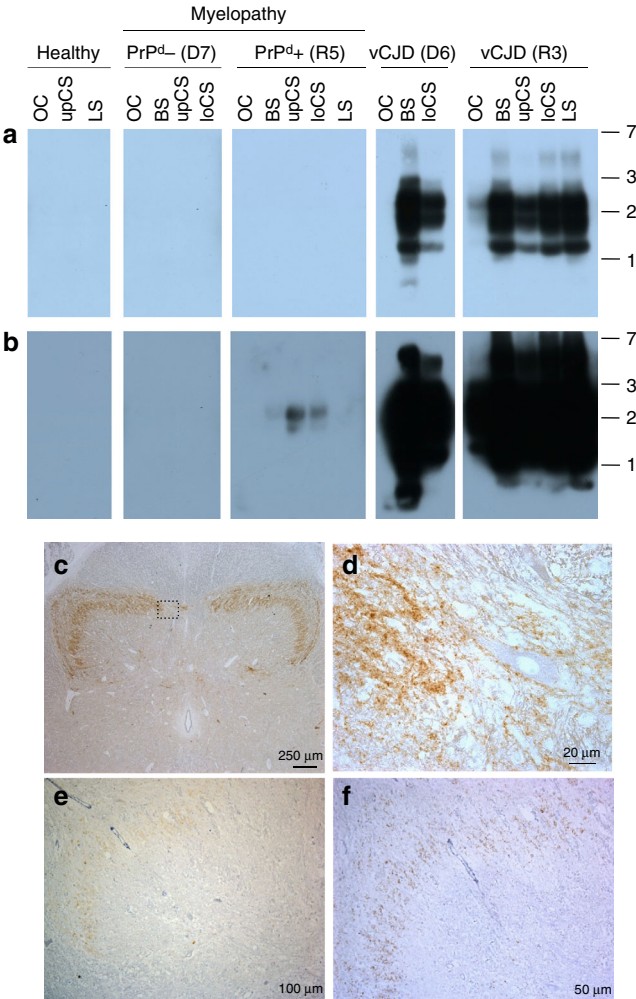

**Fig. 5** Detection of PrP in R5 myelopathic primate. Detection of PK-resistant abnormal PrP in myelopathic primates D7 and R5, and vCJD primates D6 and R3 by western blot analysis with Sha-31 anti-PrP antibody, after short-term (**a**) or long-term (**b**) exposure. OC occipital cortex, BS brainstem, upCS upper cervical cord; loCS lower cervical cord, LS lumbar cord. **c** Detection of abnormal PrPd deposits in spinal cord (12th thoracic level) of R5 primate with Sha-31 monoclonal antibody (50×). **d** Higher magnification of **c** showing perineuronal and synaptic PrP immunostaining (630×). **e** Detection of abnormal PrPd deposits in spinal cord (4th cervical level) of R5 primate with POM-1 monoclonal antibody (100×). **f** Detection of abnormal PrPd deposits in spinal cord (4th cervical level) of R5 primate with 12F10 monoclonal antibody (200×)

plasma. Indeed, crude brain homogenates induced the expected vCJD phenotypes in all macaque recipients (10/10 IC and 4/4 IV exposed animals), whereas myelopathy occurred in only one recipient (10%) of the corresponding soluble extracts (Fig. 8). Among macaque recipients of highly infectious blood products (R1–R5 and R15–R17), vCJD occurred in 3/3 recipients of whole blood and 2/3 recipients of red blood cell concentrates (RBCC) that still contained plasma and leucocytes (the third recipient developed a myelopathy with PrPd, that may be considered as an intermediate disease), whereas the two recipients of deleukocyted RBCC (only red cells and plasma) developed myelopathy. In addition, the deleukocytation of blood with low infectivity did not impair the transmission of myelopathy (macaques R8–R11).

In mice, the proportions of PrPres negative phenotypes were higher in recipients of plasma than in the recipients of buffy coat, and also in recipients of supernatants than in those animals that received pellets of soluble brain homogenate, whereas all recipients of vCJD-infected crude brain homogenates developed PrPres positive prion disease (Fig. 9a; Supplementary Table 5). Studies on soluble brain extracts were performed in two conventional mice models (Swiss and C57Bl/6 mice) and with two different infectious sources (BSE-infected cattle and primate). In those four models taken together, a complete vCJD presentation was observed in only 41% (28/69) of recipients, while 29% (20/69) exhibited incomplete syndromes. Sixteen animals exhibited the cerebral profile without PrPd while four animals developed the BS profile with cerebral and spinal PrPd. In more detail, the incidence of neurological signs and vCJD phenotypes was higher in Swiss mice than in C57Bl/6 mice; the incidence was also higher with extracts of non-human primate brain than with extracts of cattle brain and higher with pellet samples than with samples of supernatant. Those groups with higher incidence also tended to promote BS profiles, whereas the opposite groups tended to promote cerebral (C) profiles (Supplementary Table 7). No recipient of brain fractions exhibited the S profile.

**Peripheral replication pathways select PrPres negative prion**. In both experimental models, all the PrPres positive animals exhibited accumulation of PrPd in follicles in the spleen, whereas no animal developing a PrPres negative disease (NL, B, C or S phenotypes in mice, myelopathy in non-human primates) had detectable accumulation of PrPd in their lymphoid organs. We hypothesise that two distinct pathophysiological pathways may coexist (Fig. 9b). Within the classical pathway, associated with aggregated abnormal PrP, peripheral replication occurs at the level of follicular dendritic cells (FDC) in lymphoid organs, and the resulting PrPres positive disease affects the whole CNS following neuroinvasion along afferent nerves. Conversely, infectivity in a soluble form would follow an alternative pathway: peripheral replication would occur at different sites of replication that remain to be defined and lead to PrPres negative disease affecting brain or spinal cord.

## Discussion

The observed prevalence in human populations of primary and secondary (transfusion-transmitted) vCJD clinical cases is very low. This is in contrast to recent estimates suggesting, paradoxically, that there has been massive exposure of the human population[2] and that healthy carriers are close to 200-fold greater in number than the reported cases[7]. Thus, among the 10 million humans exposed in the UK, <0.5% of people would have been infected, and among these, <1% will develop classical vCJD. Here we show that exposure to blood infectivity resulted in the transmission of vCJD to only a small proportion of recipients in two independent susceptible rodent and non-human primate models. Furthermore, all recipients that developed vCJD were exposed to non-deleukocyted blood products with an expected high level of infectivity. These experimental models, therefore, replicate the situation in the human population regarding the risk of blood transfusion. No vCJD was observed in the recipients of leukodepleted blood, which underlines the protection provided by leukodepletion that is now systematically applied to human blood transfusion.

However, in the same experiments, higher proportions (2- to 7-fold more) of recipients developed other fatal neurological impairments, some of them exclusively involving the spinal cord. These unexpected neurological impairments would escape the current criteria for the diagnosis of prion diseases; such criteria include the presence of abnormal PK-resistant PrP associated with spongiform change in the brain (Supplementary Table 8). Nevertheless, we consider that these disorders have a prion

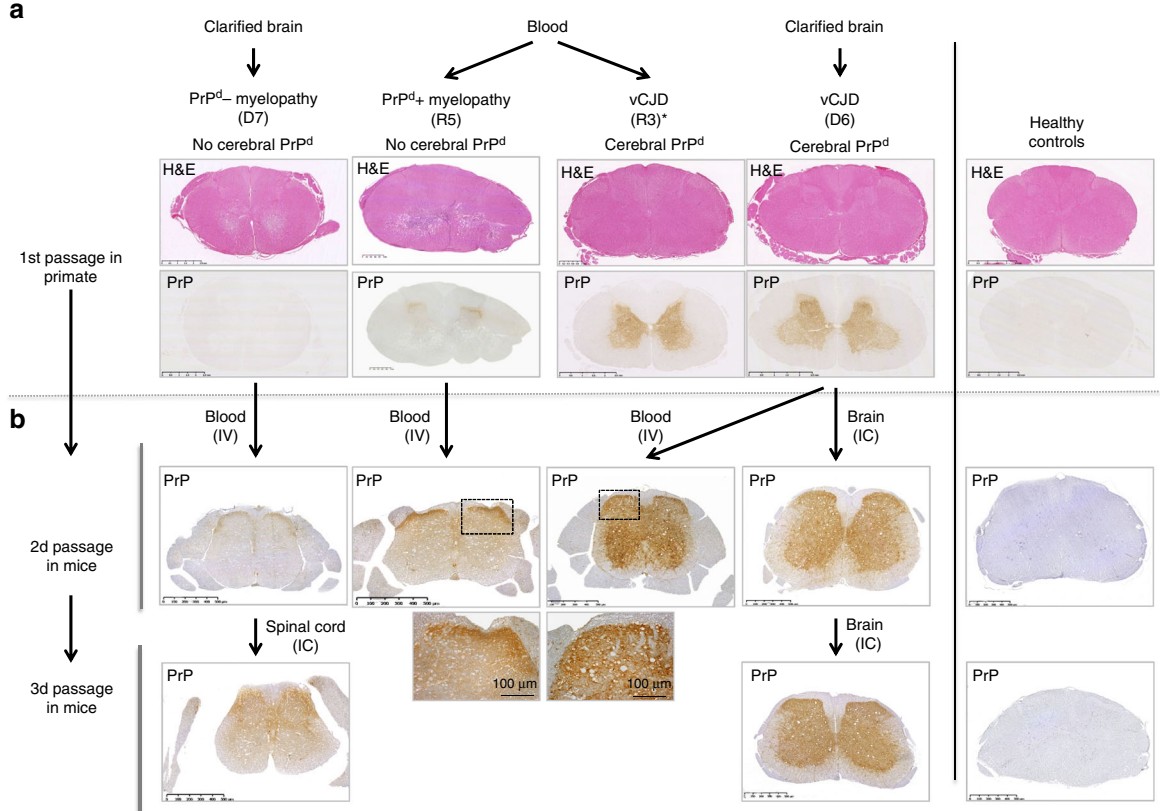

**Fig. 6** Serial transmissions of myelopathy and vCJD from macaques to mice. **a** Primates exposed to blood or cerebral products derived from vCJD-infected macaques developed vCJD (primates R3 and D6) or myelopathy (primates R5 and D7, Supplementary Text 1). Interestingly primate R3 (duplicate of primate R5) developed vCJD with spongiform changes and accumulation of PrP$^d$ in its brain and spinal cord, but it also exhibited myelopathic lesions (pseudo-necrotic lesions in the brainstem and demyelination of posterior tracts) and limited amounts of PrP$^d$ in its brain (around 10-fold lower than the other vCJD primates). **b** Corresponding blood and brain samples were inoculated intravenously (IV) or intracerebrally (IC) into Swiss mice. The lesions observed in spinal cords of macaques are presented (hemalun–eosine, H&E). The presence of PrP$^d$ in the spinal cords was detected by immunohistochemistry with different monoclonal anti-PrP antibodies (here Sha-31, for other antibodies see Fig. 5) after proteolysis treatment

aetiology, as they transmit diseases to recipient mice that exhibit the classical features of transmissible spongiform encephalopathies, i.e., accumulation of (PK-resistant or not) PrP$^d$ and spongiform change in all or part of the CNS. Similar results were achieved by intravenous administration of soluble material derived from infectious brains, which is an acknowledged model of blood infectivity. Furthermore, it is important to note that the transmission of this atypical prion disease in non-human primates is apparently not prevented by the compulsory leukodepletion step for infected blood.

Interestingly, these atypical prion phenotypes occur after incubation periods similar to vCJD in mice, and even shorter in macaques, suggesting that they are due to different variants. According to our results, the onset of such atypical prion phenotypes, including spinal cord involvement, seemed to be preferentially promoted by the administration of soluble infectivity through the intravenous route, whereas the most commonly used method (administration of crude homogenates via an intracerebral route) is associated with the classical vCJD phenotype. So, we hypothesise that non-protease-resistant soluble infectivity in blood might follow an alternate pathophysiological pathway from the stage of peripheral replication; this pathway, that may even correspond to a direct neuroinvasion as recently described[18], remains to be elucidated. Indeed, it has been shown that, in spleen, FDC trap particulate antigens but not soluble antigens of a size similar to PrP monomers[19]. Prion variants dissociated from PrP$^{res}$ could be selected through such an alternate pathway. The

commonly held opinion is that there is a unique signature for the BSE/vCJD prion, which is known to be remarkably stable even after multiple transmissions between species[20,21]. Reality might be much more complex: the emerging view is that each prion strain exists much like a quasi-species as described for viral and bacterial pathogens. According to this cloud hypothesis[22], prions would constitute a dynamic ensemble of different conformations of abnormal PrP (PrP$^d$). Some conformations would correspond to forms of PrP detectable on the basis of their resistance to proteolysis, so called PrP$^{res}$, whereas other conformations, that are oligomeric and until now undetectable by biochemical techniques, would constitute the most toxic and infectious entities. An even greater complexity may arise from the capacity of cellular PrP to interact naturally with several molecules including non-coding RNA, DNA or Aβ[23–25].

In this hypothesis, PrP$^{res}$ and infectivity/toxicity can be dissociated[26–31]. We previously reported the first obvious dissociation following primary transmission of BSE to mice[11]. Following intracerebral inoculation, all animals developed similar neurological signs but more than 50% had no detectable PrP$^{res}$. On secondary and tertiary transmissions, however, the proportion of PrP$^{res}$ positive animals gradually increased to almost 100%. Recent communications suggest that a similar situation might exist in other models of experimental exposure to prions involving swine[32] and cattle[33]. In the present study, we used two animal models and blood transfusion to demonstrate a complete dissociation that could be maintained on successive passages. The

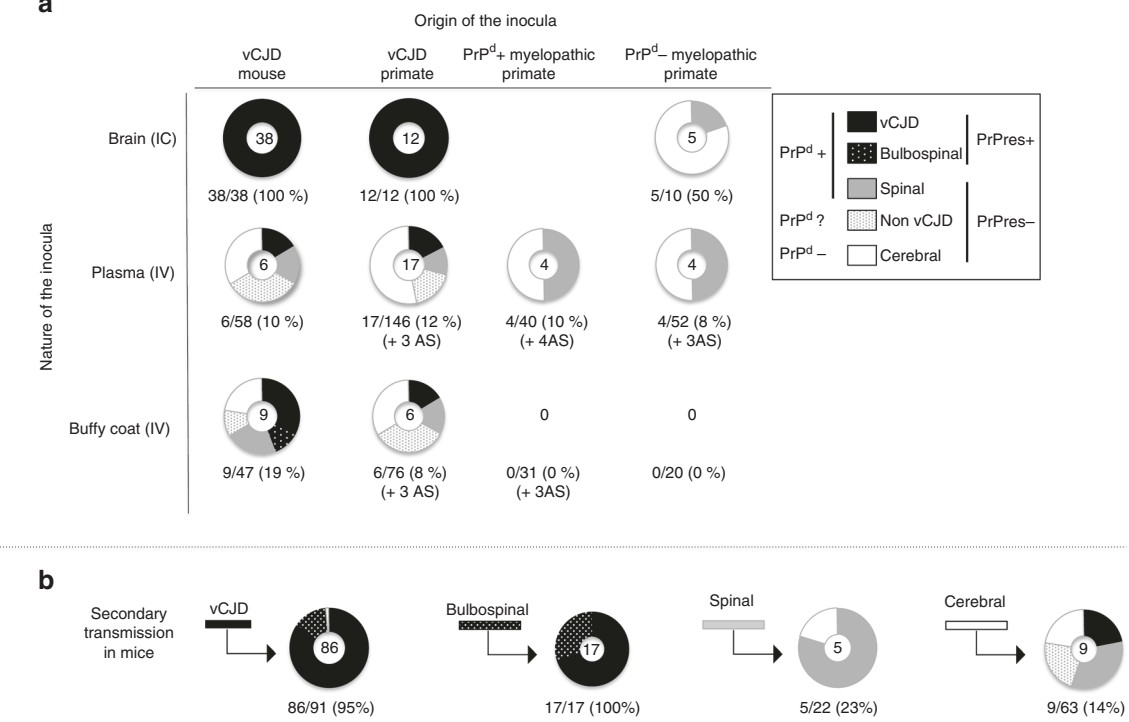

**Fig. 7** Transmission in mice. The pattern of distribution of disease profiles depending on inocula **a** after exposure to crude brain, plasma or buffy coat derived from vCJD-infected mice or primates or (PrP$^d$ negative or PrP$^d$ positive) myelopathic primates, or **b** after exposure to SNC or spleen samples derived from mice exhibiting complete or incomplete vCJD phenotypes (secondary transmission). For each inoculum, the pattern of disease profiles observed in affected mice (numbers at the centre of each circle) is depicted in five categories grouped according to their status towards abnormal PrP: classical vCJD pattern (protease-resistant PrP$^d$, black), bulbospinal (BS) pattern (protease-resistant PrP$^d$, white-spotted black), spinal (S) profile (protease-sensitive PrP$^d$, grey) or cerebral involvement grouping cerebral (C), bulbar (B) profiles and animals with only neuronal lesions (NL) (no detectable PrP$^d$, white). The fifth category (grey-spotted white) corresponds to the non-vCJD mice (affected mice devoid of PrP$^{res}$ but not sampled for histology. They might present either a spinal profile or cerebral involvement). Below each circular panel are specified the percentage of transmission and the numbers of aging spinal animals (AS)

panel of incomplete syndromes that we have observed in both mice and non-human primates suggests that the phenotypic expression of vCJD prion infection may vary. The recent description of vCJD in codon 129 heterozygous patients[34] may even expand this possible spectrum of clinical heterogeneity. Such variation ranges from the status of healthy carrier to classical vCJD and includes clinical-pathological presentations that would be excluded from the diagnosis of prion disease according to current criteria.

The complete dissociation that we observed here, with toxicity in the absence of PrP$^{res}$, counterbalances the observations made using PrP$^{res}$ amplification techniques. Such PrP$^{res}$ amplification techniques generate large amounts of PrP$^{res}$ but little or none of the toxic and infectious form of PrP. RT-QuIC is reported to be able to amplify over $10^{12}$ the quantity of PrP$^{res}$ in scrapie samples by converting recombinant cellular PrP (PrP$^c$) produced in bacteria, but almost no infectivity is generated[35]. PMCA, which uses normal brain as a substrate for the conversion of PrP$^c$, is able to maintain infectivity during successive cycles despite serial dilutions and can even generate infectivity de novo from normal mouse and hamster brains[36,37]. However up to more than 99% of the PrP$^{res}$ generated, de novo is not infectious (dissociation of at least two logs between infectivity and de novo generated PrP$^{res}$)[38]. None of these techniques, which are increasingly used for the diagnosis of human prion diseases, was able to generate PrP$^{res}$ from our infectious PrP$^{res}$ negative samples.

The neurological impairment described here in macaques appears to be unique as it has not been reported previously.

However, Holznagel et al.[39] reported a specific pattern of spinal cord involvement in cynomolgus macaques exposed orally to BSE. These animals remained asymptomatic, but they exhibited atypical deposition of PrP$^d$ that was transmissible to bovinized mice[39]. Necrotic myelopathies have been described in humans that share clinical and pathological features with the primates in the present study[40]. The neuropathological features in macaques (lesions of both the cervical anterior horns and the spinal nuclei of the trigeminal nerves) resembled focal forms of subacute necrotising encephalomyelopathy as previously reported in Leigh syndrome[41] but their posterior columns showed tract degeneration. However, as in Leigh syndrome, dysfunction of energy metabolism may be suspected[42]. Moreover, the atypical myelopathy in macaques shows similarities to other human myelopathies that range from neuromyelitis optica spectrum disorders (NMOSD), for which inclusion criteria are wide[43], to the more recently described FOSMN[44] and to certain forms of amyotrophic lateral sclerosis like FLAIL arm syndrome[45]. This latter syndrome presents as a pure lower motor neuron disorder that then evolves after a number of years to include upper motor neuron lesions[46], a feature that we did not observe in our primates. However, in the absence of suitable nursing facilities, these primates were killed relatively early for ethical reasons.

This report strongly suggests that prions might remain hidden in the population and as the majority of healthy carriers may never develop a prion disease, an even greater proportion of contaminated individuals may never be diagnosed as healthy carriers, currently diagnosed according to the presence of PrP$^d$ in

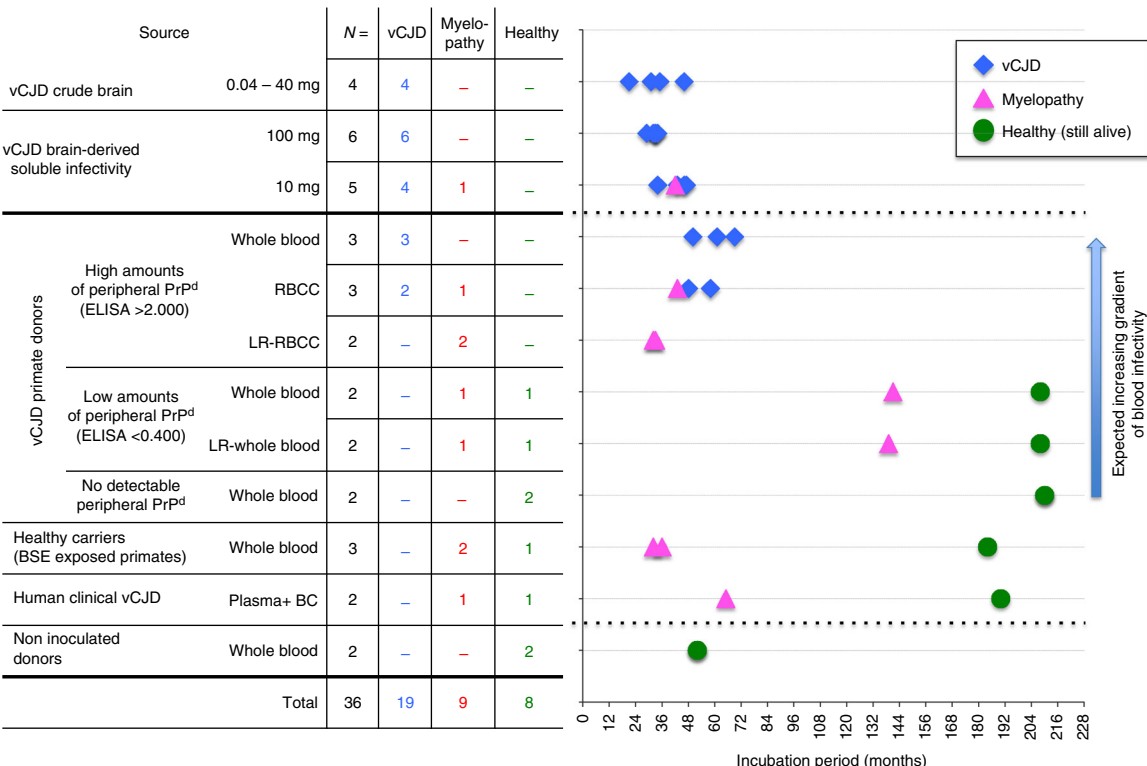

**Fig. 8** Classification of recipient primates depending upon the blood inocula. Primates were exposed through the intravenous route to whole blood or red blood cell concentrates (RBCC), deleukocyted (LR) or not (each time the equivalent of 40 ml of whole blood). The blood products were classified according to the species (primate/human) and the status with regard to peripheral vCJD replication (no, low or high levels of peripheral PrP[res], or unknown status in subclinical donors depicted here as healthy carriers). The recipient animals are categorised according to the disease phenotypes they developed and their incubation times

their lymphoid tissues. Moreover, should such atypical agents or their subsequent passages in humans lead to neurological impairment, there is a significant risk that they would not be diagnosed as related to prion infection in the absence of detectable PrP[res] and a quasi-exclusive spinal involvement. Our results enlarge the range of prion diseases that is already no longer restricted to PrP[res] positive diseases that target the brain[47–49].

In conclusion, the range of incomplete syndromes that we observed between healthy carriers and typical vCJD indicates that multiple forms of prion variants can coexist and may emerge in different forms depending upon the conditions under which transmission occurred. This has obvious consequences for public health, and questions the uniqueness of the BSE/vCJD strain[50] and our capacity to detect and prevent all infectious forms of prion disease.

## Methods

**Ethics statement**. Primates and mice were housed and handled in accordance with the European Directive 2010/63 related to animal protection and welfare in research, and were under constant internal surveillance by veterinarians. Social enrichment was a constant priority with individual activities and feeding controlled according to the risk of infection. Animals were handled under anaesthesia to limit stress, and killing was performed for ethical reasons when animals lost autonomy. CETEA ethical committee approved the present experiments (approvals 12-020, 12-067, 12-068, 12-070, 12-072 and 14-072). Samples derived from patients were obtained in accordance to the rules in force at that time. The sizes of experimental groups were defined in accordance with the reduction and refinement concepts of 3R's, depending on the respective expected rates of transmission. According to the specificity of the macaque model, each case is to be considered as single clinical case.

**Experimental animals**. Captive-bred 2–5-year-old male cynomolgus macaques (*Macaca fascicularis*) were provided by Noveprim (Mauritius), checked for the absence of common primate pathogens before importation, and handled in

accordance with national guidelines. They were all Met/Met homozygous at codon 129 of the prion protein gene (*PRNP*). Six weeks old, female C57Bl/6N and Swiss ND4 mice were provided by Janvier Labs (France). Animals were housed in level-3 animal care facilities (agreement numbers A 92-032-02 for animal care facilities, 92–189 for animal experimentation) and regularly examined at least once a week. No statistical randomisation method was used to allocate animals to experimental groups. Samples for biochemical and histological analyses were coded (two different codes) and respective investigators were blind during analyses.

**Inocula**. All the brain inocula were 1 or 10% (weight/volume) homogenates in 5% glucose solution. Blood products were sampled in citrate buffer.

**Brain sample preparations in mouse studies**. The inocula used for the experiments using soluble brain infectivity were brain inoculum from BSE-infected cattle (a mixture of 11 cattle brains[51]) and brain inoculum from a BSE-infected primate (a primate that was injected intracerebrally with 0.5 μg of this BSE inoculum, and developed BSE 85 months post inoculation). Corresponding high-speed supernatant (S[HS]) and pellet (P[HS]) were prepared according to the protocol described by Berardi et al.[17] to model soluble and microsomal blood infectivity respectively. Briefly, 10% brain homogenates were centrifuged (low speed) at 825×*g* for 15 min. The resulting supernatant was ultracentrifuged (188,000×*g* for 1 h) after extensive sonication. Corresponding supernatant (S[HS]) and pellet (P[HS]) fractions were inoculated IV through the tail vein with the equivalent of 1.5 mg of brain (200 μl) per animal. As controls, mice were inoculated intracerebrally with a crude 10% (w/vol) homogenate derived from the same BSE-infected primate brain (20 μl, equivalent to 2 mg of brain).

**Brain sample preparations in primate studies**. In the first group of animals exposed to crude brain homogenates, four primates were inoculated IV with serial dilutions of a crude brain homogenate derived from a BSE-infected primate (primate D8)[52]. For the second group of animals exposed to soluble brain infectivity, clarified brain inocula were obtained by centrifugation at 1500×*g* for 10 min after extensive sonication of brain inocula derived from two infected primates (primate D14 orally exposed to 5 g of BSE-infected cattle brain[16], and primate D1 infected by intracerebral injection with human vCJD[20]). The supernatants obtained were injected into recipient macaques (equivalent to 10 or 100 mg of brain)[53].

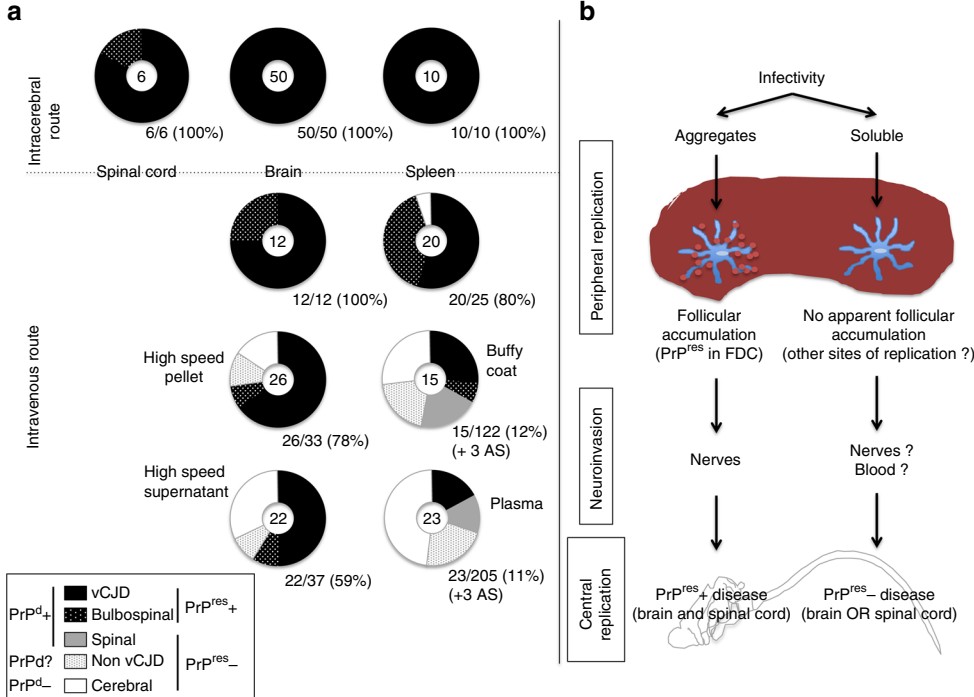

**Fig. 9 a** The pattern of distribution of disease profiles in mice according to the kind of inocula (brain, soluble infectivity or blood products). For each inoculum, the pattern of disease profiles observed in affected mice (numbers at the centre of each circle) is depicted in five categories grouped according to their status towards abnormal PrP: classical vCJD pattern (protease-resistant PrP[d], black), bulbospinal (BS) pattern (protease-resistant PrP[d], white-spotted black), spinal (S) profile (protease-sensitive PrP[d], grey) or cerebral involvement grouping cerebral (C), bulbar (B) profiles and animals with only neuronal lesions (NL) (no detectable PrP[d], white). The fifth category (grey-spotted white) corresponds to the non-vCJD mice (affected mice devoid of PrP[res] but not sampled for histology. They might present either spinal profile or cerebral involvement). Below each circular panel are specified the percentage of transmission and the numbers of aging spinal animals (AS). **b** Two different putative pathophysiological pathways may occur depending on the aggregated or soluble nature of the initial infectivity

**Blood sample preparations**. Blood samples from three UK and one French vCJD patients were used in parallel with primate blood samples drawn into sodium citrate. Corresponding RBCC, plasma and buffy coat fractions were collected after centrifugation at $2000 \times g$ for 13 min. Deleukocyted blood samples were prepared by pooling RBCC and plasma after fractionation. Deleukocyted RBCC were obtained through filtration on a commercial device (LST2, MacoPharma, $<1 \times 10^6$ leuco-cytes/unit after filtration) and then re-suspended in corresponding plasmas in proportions used for paediatric transfusion[53].

Sixty Swiss mice were inoculated intracerebrally with 2 mg of brain from Swiss mice infected with the vCJD-adapted strain. Their blood was collected 20 weeks post inoculation, pooled and centrifuged at $2000 \times g$ for 13 min. Buffy coat and plasma were recovered after fractionation.

**Inoculation of blood products**. Transfusion experiments in primates are extensively described in the Supplementary Text 1. Primates were injected IV with volumes of blood products equivalent to 40 ml of whole blood unless otherwise stated. Mice were inoculated IV with 0.2 ml of plasma or buffy coat (equivalent to 0.5 and 1 ml of whole blood, respectively) derived from either vCJD-infected mice, vCJD-infected primates (D3, D4, D5, D6, D19 and R15) or myelopathic primates D7 and R5 that developed the myelopathic syndrome without or with accumulation of PrP[d] in their spinal cord, respectively.

**Transmission experiments**. C57Bl/6 and Swiss mice were inoculated intracerebrally (2 mg of tissue) or IV (0.2 mg of tissue) with brain, spinal cord or spleen homogenates from mice or primates.

**Neuropathology and immunohistochemistry**. Tissues were fixed in 4% formalin for histological examination. Neuropathology and immunohistochemical detection of PrP[d] were performed on brain sections as previously described using 3F4, Sha-31, SAF-60, POM-1, SAF-32 or 12F10 antibodies after a proteolysis step[54]. Immunohistochemical detection of GFAP was performed using polyclonal rabbit anti-GFAP antibody (Dako); apoptotic cells were detected using the In Situ Cell Death Fluorescence Kit (Roche)[55].

**Biochemical PrP analysis**. PrP was purified according to the TeSeE purification protocol (Bio-Rad), in the presence (40 µg/mg protein) or absence of proteinase K.

Purified samples were detected by ELISA (TeSeE Kit for primate samples[53], or a specific ELISA using Saf-53 as the capture antibody and 11C6 as the detection antibody for mouse samples[56]), or processed for western blot analysis as previously described[54] using 3F4, Saf-37 or Sha-31 anti-PrP monoclonal antibodies.

**PMCA reactions**. A transgenic mouse line that expresses ovine $A_{136}R_{154}Q_{171}$ PrP variant PrP[c] (tgShXI) was used to prepare the PMCA substrate as previously described[57,58]. PMCA amplification was performed as previously described[59]. Each PMCA run included a reference vCJD sample (10% brain homogenate) as a control for the amplification efficiency. Unseeded controls (one unseeded control for eight seeded reactions) were also included in each run. PMCA reactions were seeded with 1/50 diluted tissues homogenates (10% weight/volume). For each sample, at least four replicates were tested in two independent runs. Control sample was serially diluted 10-fold ($10^{-2}$–$10^{-10}$) before being used to seed PMCA reactions.

Samples were submitted to five successive amplification rounds, each composed of 96 cycles (10 s sonication-14 min and 50 s incubation at 39.5 °C) in a Qsonica700. After each round, (i) reaction products (one volume) were mixed with fresh substrate (nine volumes) to seed the following round while (ii) a part of the same product was analysed by western blot (WB) for the presence of PK-resistant PrP as previously described[59]. Immunodetection was performed using two different monoclonal PrP-specific antibodies, Sha-31 (1 µg/ml)[60], and 12B2 (4 µg/ml)[61], which recognise the amino-acid sequences YEDRYYRE (145–152), and WGQGG (89–93), respectively.

**RT-QuIC**. RT-QuIC was performed as originally described[62]. The buffer contained 400 mM NaCl and Syrian golden hamster recombinant PrP (aa 23–231) was used as substrate. Dilutions of brain and spinal cord homogenates were prepared in N2 buffer and each sample was run in triplicate. Reactions were seeded with 2 µl of dilution in a final volume of 100 µl. Plates were incubated in a BMG Fluostar Omega plate reader at 45 °C for 70 h, with cycles of 1 min shake (600 rpm double orbital) and 1 min rest throughout the incubation.

**MRI analysis**. Magnetic resonance imaging (MRI) was conducted on a 7 Tesla horizontal system (Varian). Animal positioning and monitoring was performed as previously described[63] using a dedicated suit to maintain level-3 confinement.

**Data availability**. The data that support the findings of this study are available from the corresponding author upon reasonable request.

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

## Acknowledgements

We specially thank D. Jouy for her contribution to this work, C. Lasmezas and C. Jas-Duval for the initiation of the animal experiments, Prof A. Aguzzi for the generous gift of POM-1 antibody and N. Lescoutra, V. Durand, S. Luccantoni, E. Correia, C. Dehen, A. Culeux, C. Durand, S. Jacquin, L. Sourd and J. Delmotte for precious technical assistance. We also thank Drs. S. Blot, J.M. Helies, C. Joubert, J. Ironside, H. Budka, P.L. Gambetti, P. Brown, D. Seilhan, R. Grosse, M. Pocchiari, P. Brandel and L. Court for their expert advice about clinical and histological observations of the primates. We specially thank Prof R. O. Weller for editing the manuscript. We acknowledge European Commission (QLK1-CT-2002-01096), French Research Funding Agency (ANR), Health Canada and MacoPharma for funding parts of those experiments.

## Author contributions

E.E.C. coordinated the study, designed the mouse experiments, contributed to the biochemical and histological studies, analysed the data and wrote the manuscript. J.M. performed all the histological analyses and scoring. N.J. contributed to the animal experiments. V.L. performed the MRI analyses. E.L. performed the QuiC analyses. N.S. contributed to the analysis of the data. C.S. contributed to the design of primate experiments and editing of the manuscript. A.P.-L. performed the CSF analyses. M.E. supervised the deep sequencing studies. O.A. performed the PMCA analyses. S.H. contributed to the QuiC analyses, discussed the data and edited the manuscript. P.H. contributed to the MRI analyses. J.-P.D. supervised the study, contributed to the experimental design, optimised the immunohistochemical analyses, analysed the data and wrote the manuscript.

## Additional information

**Competing interests:** The authors declare no competing financial interests.

