## [Peer review file · Nature Communications]

Reviewers' comments:

Reviewer #1 (Remarks to the Author):

The manuscript by Emmanuel Comoy and colleagues describes that intravenous transfusion of variant CJD-infected blood into macaques and mice can result in generation of an atypical new class of prion diseases. In order to test the risk of infection with variant Creutzfeldt-Jacob disease (vCJD) prions upon blood transfusion, the authors transfused blood products from symptomatic and non-symptomatic vCJD patients into non-human primates (*Cynomolgus* macaques) and wild-type mice, in a very extensive study. The authors found that they can infect both mice and monkeys with vCJD derived from blood via transfusion. Interestingly, only a fraction of infected animals developed vCJD. Very unexpectedly, in both species an atypical prion disease was produced, lacking classical biochemical and pathological markers. In macaques this novel prion disease manifested as a fatal myelopathy, which was transmissible into mice. Overall, the authors did a very careful and well controlled analysis and they provide important data which are novel, highly significant, and which have direct public health significance. These data extend the growing spectrum of atypical prion diseases. Importantly, the unexpected clinical presentation in the non-human primate model strongly suggests that current diagnostic criteria might not detect related atypical cases in humans. The manuscript is well done, experiments are clearly described and well controlled, and conclusions are justified by the experimental data. This referee does not see any major flaws. The differentiation into "endogenous" and "exogenous" infectivity is somewhat misleading. Why not describing it as spiked (brain) and non-spiked (blood) material? The scheme in Fig. 6b is interesting (aggregates vs. soluble), but there are alternative explanations which the authors should take into account. One is that they are looking at a titer effect, associated with detectable PrP^{Sc} or not, which would not be correlated to a different site of prion replication. Their most sensitive technique, RT-QuIC, was used only for brain and spinal cord samples, apparently not for spleen or lymph nodes. There is an argument against this scenario which should be used by the authors. The incubation time for primates developing myelopathy (Fig. 5) is very short, even shorter than for the vCJD phenotype. This intriguing observation needs more discussion. Given the potential high impact for the human prion field and the public health implications, it is very important that these data are made available for the field.

Minor points:

- 1) Spleen data are not shown. Is there a reason why?
- 2) Codon 129 in humans: Given the very recent report that vCJD now shows up in codon 129 heterozygous patients, this expands the possible spectrum of clinical heterogeneity. The authors might want to discuss this on the background of their data.
- 3) Immunoblot Fig. S5: -PK signals are weak, so a conclusion on +PK signals is difficult.
- 4) RT-QuIC Fig S6: Legends in figure (insert) is not readable. Was only a 10⁻³ dilution used for the samples? Lower dilutions might yield a positive result. Were PrP substrates of other species used (e.g. macaque)?

Reviewer #2 (Remarks to the Author):

This is a very long and interesting study describing the appearance of some unexpected clinical phenotypes in primates or transgenic mice transfused with blood products from symptomatic and non-symptomatic infected donors. While the majority of the animals developed the expected vCJD phenotype, others showed a unique class of neurological diseases. These disorders can be transmitted in a second passage to mice with a pathognomonic accumulation of abnormal prion protein. These findings may expand the spectrum of prion diseases and suggest that we might be underestimating the number of prion infections produced as a consequence of the BSE/vCJD epidemic.

The main message of this study is very important and the experiments are in general well done.

The article reports an impressive collection of experiments which have taken a tremendous amount of time and resources to complete. However, because of the very large amount of data and the rather complicated way the authors composed the article, it is extremely confusing and difficult to follow. In its present form, the manuscript consists of 6 main figures, 10 supplementary figures and 9 supplementary tables. I will advise the authors to simplify the manuscript, perhaps removing some of the data (which can be published in a separated article) and providing a more direct message with the most essential data.

In addition I recommend to address the following issues:

1. An important issue which is not dealt appropriately in the paper is a definitive demonstration of the absence of PrPd in some of the animals that show the incomplete neurological phenotype. Although it is entirely possible that PrPD is not present at all, this is unlikely since the material was infectious in a subsequent round of infection and produced PrPres material. The authors claim they did PMCA and RT-QuIC to attempt detecting traces of the abnormal protein. However, only the RT-QuIC data is shown. RT-QuIC is known to be inefficient for detecting vCJD prions. At the contrary, PMCA detects vCJD with very high efficiency. The PMCA data should be shown along with a clear demonstration that the PMCA assay is operating at its maximum efficiency, as described recently in a couple of articles published in Science Translational Medicine.

2. Authors show data of i.v. infection using brain preparations. How did they avoid mortality produced by pieces of brain extracts? It is well known that i.v. infection with tissue extracts is difficult, leading to strokes in many of the inoculated animals.

3. In page 7, it is stated that some of the animals were not analyzed by histology; why is that? do they still have the tissue? if so, histological analysis should be done with all animals.

4. In page 15, they state that 99% of PrPres generated material from PMCA is not infectious and they cite for this an article from Chesebro's group. Although, this is correct, it appears this only applies to the hamster 263K strain. Various other studies from diverse labs (Soto, Bartz, Supattapone) have shown that the infectivity titer of PMCA generated material is the same as the in vivo produced prions.

5. Do the authors have an explanation of why these new phenotypes have not been detected before. Many people have done injection of vCJD into primates or transgenic mice, always showing the typical vCJD manifestation.

Reviewer #3 (Remarks to the Author):

Comoy and colleagues have presented the first experimental evidence that transfusion of blood products from symptomatic and non-symptomatic infected donors in mice and primates induces vCJD and also a unique class of neurological diseases. These disorders can all be retransmitted to mice with a pathognomonic accumulation of abnormal prion protein, thus expanding the spectrum of prion diseases. The authors suggest that the intravenous route promotes propagation of masked prion variants according to different mechanisms involved in peripheral replication. They state that the impact of such variants on human health cannot be estimated by epidemiological surveys relying upon current diagnostic criteria for vCJD and so they suggest that these criteria should be adjusted accordingly. They conclude that the range of incomplete syndromes that they observed between healthy carriers and typical vCJD indicates that multiple forms of prion variants can coexist and may emerge in different forms depending upon the conditions under which transmission occurred.

Overall this is a carefully written manuscript with suitable statistics and well-presented data in the main body of the text. The main conclusion from the manuscript that "multiple forms of prion variants can coexist and may emerge in different forms depending upon the conditions under

which transmission occurred" is not a new phenomenon. Other groups have stated this in the past and prion biologists have been aware of this for a while. Having said that, there are some new and very interesting findings in this current manuscript.

There is however a need to shorten the amount of supplementary data that supports this paper. It is not necessary or useful to have 35 pages of supplementary data and less pages in the actual document to be published. I do feel there is a need to shorten and condense the amount of supplementary information so that the manuscript is more self-contained and less cumbersome.

Major point: The manuscript is interesting and has thrown up some important information regarding incomplete syndromes that require further analysis and study in order to clarify some of the points raised by the authors. I agree there are obvious consequences for public health and there is a need for this type of work to highlight the issues.

I appreciate the vast amount of work involved and the fact that prion studies can take up a huge amount of time (especially primate studies). Some of the evidence that has been given is inconclusive and so requires further analysis. Much of the content comes over as not being complete or needing a bit more experimentation or analysis in order to generate conclusive information.

Minor point: Numbering lines from 1 on every page is actually not helpful.

I am slightly disturbed by the word ".....considered to be..." such as in lines 21 and 25 (page 6).

This does not demonstrate a definite result and so it needs further study.

Every now and then there are phrases that make me slightly nervous of the robustness of the data shown.

I have an issue with what the authors have said on page 7 lines 24 ad 25. To say an animal "might exhibit...." is hardly scientific and needs much better explanation or clarification.

I feel the paper needs condensing by removing some of the supplementary data and text and it needs some clarification in places where animals were not tested by one or more protocols and were then "considered to be....". The samples either need to be tested fully or the text needs adjusting to make things clearer, less ambiguous and more scientific.

TITLE OF MANUSCRIPT: Transfusion of variant CJD-infected blood reveals a novel class of prion disorders

MANUSCRIPT NUMBER: NCOMMS-17-00355A

CORRESPONDING AUTHORS NAME: Comoy Emmanuel

Answers to Reviewers' comments:

Reviewer #1 (Remarks to the Author):

The manuscript by Emmanuel Comoy and colleagues describes that intravenous transfusion of variant CJD-infected blood into macaques and mice can result in generation of an atypical new class of prion diseases. In order to test the risk of infection with variant Creutzfeldt-Jacob disease (vCJD) prions upon blood transfusion, the authors transfused blood products from symptomatic and non-symptomatic vCJD patients into non-human primates (cynomolgus macaques) and wild-type mice, in a very extensive study. The authors found that they can infect both mice and monkeys with vCJD derived from blood via transfusion. Interestingly, only a fraction of infected animals developed vCJD. Very unexpectedly, in both species an atypical prion disease was produced, lacking classical biochemical and pathological markers. In macaques this novel prion disease manifested as a fatal myelopathy, which was transmissible into mice. Overall, the authors did a very careful and well controlled analysis and they provide important data which are novel, highly significant, and which have direct public health significance. These data extend the growing spectrum of atypical prion diseases. Importantly, the unexpected clinical presentation in the non-human primate model strongly suggests that current diagnostic criteria might not detect related atypical cases in humans. The manuscript is well done, experiments are clearly described and well controlled, and conclusions are justified by the experimental data. This referee does not see any major flaws. The differentiation into "endogenous" and "exogenous" infectivity is somewhat misleading. Why not describing it as spiked (brain) and non-spiked (blood) material?

We agree with the reviewer that this notion of "exogenous" and "endogenous" can be misleading. We initially used it as reminder of the terminology used in experimental models dedicated to evaluate the efficiency of blood safety processes towards prion. We thus changed the text as follows:

- **Page 4 line 14:** "...using endogenous blood infectivity ... and exogenous brain infectivity ..." by "...using blood infectivity ... and brain infectivity..."
- **Page 6 line 6:** "...mice exposed to exogenous infected brain extracts or endogenous infected blood..." by "...mice exposed to infected brain extracts or infected blood"
- **Page 11 line 14:** "...within the animals exposed to exogenous/endogenous infected blood..." by "...within the animals exposed to brain or blood infectivity..."
- **Page 13 line 5:** "...exposure to vCJD infectivity through endogenous blood..." by "...exposure to blood infectivity..."

The scheme in Fig. 6b is interesting (aggregates vs. soluble), but there are alternative explanations which the authors should take into account. One is that they are looking at a titer effect, associated with detectable PrPSc or not, which would not be correlated to a different site of prion replication. Their most sensitive technique, RT-QuIC, was used only for brain and spinal cord samples, apparently not for spleen or lymph nodes. There is an argument against this scenario which should be used by the authors. The incubation time for primates developing myelopathy (Fig. 5) is very short, even shorter than for the vCJD phenotype. This intriguing observation needs more discussion.

We agree with the reviewer that alternative explanations to our main hypothesis of different ways of peripheral replication may exist. We were indeed surprised to observe that these atypical phenotypes may occur with incubation periods similar, or

even shorter, than those of vCJD, and the only correlation that we could make for the occurrence of those atypical phenotypes was the relative proportion of soluble infectivity. We did not inject serial dilutions of the initial blood products, as we anticipated from our knowledge of the primate model that resulting incubation periods would extend for decades. However, it would be of interest to study this hypothesis of titre effect in mouse models and it should be a full study by itself.

The hypothesis of a direct relation between the emergence of atypical phenotypes and the relative proportion of soluble infectivity is moreover sustained by the recent publication of the group of Caughey (*Bett et al., Neuropathologica acta, 2017*), where they show that soluble prions have higher capacity of direct neuroinvasion than fibrillar strains. We improved the discussion section as follows: *“Interestingly these atypical prion phenotypes occur after incubation periods similar to vCJD in mice, and even shorter in primates, suggesting that they are due to different variants. According to our results.... this pathway, that may even correspond to a direct neuroinvasion as recently described¹⁸, remains to be elucidated.”*

Given the potential high impact for the human prion field and the public health implications, it is very important that these data are made available for the field.

Minor points:

1) Spleen data are not shown. Is there a reason why?

As we mentioned in the text (notably at the beginning of the paragraph “Mechanisms of peripheral replication selects PrP^{res} negative prions” in the results section and in the supplementary table 5 at the level of the description of disease profiles), PrP^{res} accumulation was detected with current techniques in the spleens of vCJD and BS animals but not in the animals exhibiting other disease phenotypes. According to the vast amount of data provided in this manuscript (as underlined by the referees), we estimated that this “black and white” situation needed less to be illustrated than the lesions and the abnormal PrP accumulation that we observed within the CNS (brain + spinal cord), and we preferred to focus on those last ones in this manuscript. However, we are ready to provide a supplementary figure illustrating this point if requested.

We are currently deeply optimizing immunohistochemical analysis with new, original techniques of epitopes retrieval. The preliminary observations suggest that intermediate situations might occur within this “black and white” situation, but this is a full long story that will need another complete manuscript to be told.

2) Codon 129 in humans: Given the very recent report that vCJD now shows up in codon 129 heterozygous patients, this expands the possible spectrum of clinical heterogeneity. The authors might want to discuss this on the background of their data.

We included the sentence *“The recent description of vCJD in codon 129 heterozygous patients may even expand this possible spectrum of clinical heterogeneity.”* within the discussion section (page 15 line 21).

3) Immunoblot Fig. S5: -PK signals are weak, so a conclusion on +PK signals is difficult.

We modified the figure (now figure 4 in the main text) by replacing the picture of the immunoblot with a higher exposure, and included immunoblot with two other

antibodies (Saf-37 and Saf-60) recognizing other epitopes of PrP.

4) RT-QuIC Fig S6: Legends in figure (insert) is not readable. Was only a 10⁻³ dilution used for the samples? Lower dilutions might yield a positive result. Were PrP substrates of other species used (e.g. macaque)?

We have modified the legend to make it more legible. Preliminary experiments showed a matrix effect at lower dilutions (10⁻¹, 10⁻²) leading to negative RT-QuIC detection of abnormal PrP in brain homogenates from macaques inoculated with vCJD. We also observed a similar effect with other prion isolates such as BSE and sCJD. The lowest dilution of brain homogenate yielding positive results is 10⁻³. This is indeed a well-known phenomenon with PMCA and QuIC techniques, for which biological samples contain inhibitors.

In our hands, RT-QuIC performed with human and hamster recombinant PrPs showed the same sensitivity to detect vCJD in macaques. We chose to use hamster PrP in our experiments because we have a larger access to this recombinant PrP and because we extensively validated its performances in CJD (more than 1000 CSF from patients with a suspected diagnosis of CJD were examined). We have no access to macaque recombinant PrP

Reviewer #2 (Remarks to the Author):

This is a very long and interesting study describing the appearance of some unexpected clinical phenotypes in primates or transgenic mice transfused with blood products from symptomatic and non-symptomatic infected donors. While the majority of the animals developed the expected vCJD phenotype, others showed a unique class of neurological diseases. These disorders can be transmitted in a second passage to mice with a pathognomonic accumulation of abnormal prion protein. These findings may expand the spectrum of prion diseases and suggest that we might be underestimating the number of prion infections produced as a consequence of the BSE/vCJD epidemic.

The main message of this study is very important and the experiments are in general well done. The article reports an impressive collection of experiments which have taken a tremendous amount of time and resources to complete. However, because of the very large amount of data and the rather complicated way the authors composed the article, it is extremely confusing and difficult to follow. In its present form, the manuscript consists of 6 main figures, 10 supplementary figures and 9 supplementary tables. I will advise the authors to simplify the manuscript, perhaps removing some of the data (which can be published in a separated article) and providing a more direct message with the most essential data.

We agree with the reviewer #2 that the amount of data is consequent (more than 1000 mice) and we confirmed that those studies took a long time. Another reviewer formulated similar comment. We lightened the supplementary information by transferring three supplementary figures, one supplementary table and one supplementary note in the main text, and merged three supplementary figures in one. Our results are based on primary exposure of primate and mice to brain and blood samples, and secondary transmission to mice. We believe that all those different independent experiments are like Hercules' columns that strengthen each other to be fully demonstrative. The only set of data that eventually might be removed are the alternative etiologies that we explored. However, these explorations were systematically required by our scientific colleagues when we discussed with them of

our observations on primates, to eliminate obvious non-prion cause to this myelopathic syndrome. We made the choice to keep these information as a supplementary note, and also the detailed experiments on primate transmission and secondary transmission on mice, that are information dedicated to specialists.

In addition I recommend to address the following issues:

1. An important issue which is not dealt appropriately in the paper is a definitive demonstration of the absence of PrPd in some of the animals that show the incomplete neurological phenotype. Although it is entirely possible that PrPD is not present at all, this is unlikely since the material was infectious in a subsequent round of infection and produced PrPres material. The authors claim they did PMCA and RT-QuIC to attempt detecting traces of the abnormal protein. However, only the RT-QuIC data is shown. RT-QuIC is known to be inefficient for detecting vCJD prions. At the contrary, PMCA detects vCJD with very high efficiency. The PMCA data should be shown along with a clear demonstration that the PMCA assay is operating at its maximum efficiency, as described recently in a couple fo articles published in Science Translational Medicine.

RT-QuIC is indeed known through publications of others to be inefficient for detecting vCJD prions, but in this manuscript and in a previous paper (Levavasseur *et al.*, *PlosOne* 2017), we showed that RT-QuIC is able to detect vCJD prions in our macaque model with similar efficiency as s-CJD. We included PMCA data in the manuscript, which show exactly the same results as RT-QuIC. These PMCA data correspond to new experiments performed by our colleagues from INRA, who previously demonstrated the high sensitivity of their PMCA technique in the macaque model since they detected preclinical blood samples (Lacroux *et al.*, *Plos Pathogens*, 2014). The results of PMCA (that confirm the previous ones), RT-Quic and conventional biochemical detection of PrPres were combined in the figure 4. Taken altogether, these data suggest that abnormal PrP exists in myelopathic animals, but it has not the properties we are used to observe with other prion strains that are: 1) protease resistance as observed with biochemical approaches, 2) the ability to generate PK resistant PrP through seeding as observed with PMCA, and 3) the ability to generate amyloid formation through seeding as observed with RT-QuIC.

2. Authors show data of i.v. infection using brain preparations. How did they avoid mortality produced by pieces of brain extracts? It is well known that i.v. infection with tissue extracts is difficult, leading to strokes in many of the inoculated animals.

The wide majority of the animals were inoculated with clarified brain preparations obtained after sonication and centrifugation as described in the materials and methods section. These preparations are devoid of the pieces of brain leading to strokes as described by the reviewer. Only four primates (D9, D10, D11 and D12) were exposed to crude brain homogenates through the intravenous route: they received either 40 mg in 4 ml (1% brain homogenate) or 4 mg, 0.4 mg or 0.04 mg in 1 ml (0.4, 0.04 or 0.004% brain homogenate). Those diluted brain homogenates were slowly injected through the intravenous route, avoiding thus thrombosis and embolism.

3. In page 7, it is stated that some of the animals were not analyzed by histology; why is that? do they still have the tissue? if so, histological analysis should be done with all animals.

We did not mention in our manuscript that some animals were “*not analyzed by histology*”, but “*not sampled for histology*”; indeed, those animals were found dead and their brain were not in appropriate status to allow pertinent histological analysis: their CNS were thus entirely sampled for biochemical analysis. The supplementary table 1 has for main purpose to detail the type of analysis performed or not for each sample. To be clearer on this point, we precised in the sup. Table 1 that “NT=not tested **because not sampled**”.

4. In page 15, they state that 99% of PrPres generated material from PMCA is not infectious and they cite for this an article from Chesebro's group. Although, this is correct, it appears this only applies to the hamster 263K strain. Various other studies from diverse labs (Soto, Bartz, Suppattapone) have shown that the infectivity titer of PMCA generated material is the same as the in vivo produced prions.

Indeed, this publication of Chesebro's group that we cited is focused on 263K strain. In the papers mentioned by the reviewer, other prion strains are described but none is v-CJD. They also described a discrepancy between PrP^{res} and infectivity but in a lower proportion (75% to 90% of PrP^{res} generated material from PMCA are not infectious in those papers). We propose to modify the sentence “up to more than 99% of the PrP^{res} ...”

5. Do the authors have an explanation of why these new phenotypes have not been detected before. Many people have done injection of vCJD into primates or transgenic mice, always showing the typical vCJD manifestation.

We agree with the reviewer that many people have done injection of vCJD into primates or transgenic mice, always showing the typical vCJD manifestation: we were notably the first group to describe the vCJD profile in macaque (*Lasmezas et al. Nature 1996*) and we reproducibly observed this phenotype in our animals exposed to infectious brain material after intracerebral or oral exposures. However, we are from our knowledge the only group that has performed contamination through blood transfusion or intravenous exposure with clarified brain in macaques, which are the two conditions of experimental exposure for which we describe myelopathy here. Concerning mice, the two models (Swiss and C57Bl/6) that we describe here are conventional mice and not transgenic mice: we chose conventional mice here to avoid the bias of selection of PrP^{res} positive strain variants by transgenic mice because they overexpress PrP. From our knowledge, nobody published transfusion studies in the experimental model of vCJD infection of conventional Swiss mice. For the C57Bl/6 lineage, this is not the first time that the onset of alternative disease phenotypes was reported after injection of vCJD: when we exposed C57Bl/6 mice to BSE through the intracerebral route in 1997 (*Lasmezas et al., Science 1997*), we observed the occurrence of PrP^{res} negative prion diseases. Like in the present manuscript, those original diseases were also transmissible with the onset of PrPres accumulation in secondary recipients. Moreover, it must be reminded that in these conventional C57Bl/6 mice, the group of James Ironside described the occurrence of a type 1 prion disease after exposure to vCJD, it is to say a disease profile affiliated to sporadic CJD and thus different from the typical vCJD manifestation (*Yull et al., American Journal of Pathology 2006*).

Reviewer #3 (Remarks to the Author):

Comoy and colleagues have presented the first experimental evidence that transfusion of blood products from symptomatic and non-symptomatic infected donors in mice and primates induces vCJD and also a unique class of neurological diseases. These disorders can all be retransmitted to mice with a pathognomonic accumulation of abnormal prion protein, thus expanding the spectrum of prion diseases. The authors suggest that the intravenous route promotes propagation of masked prion variants according to different mechanisms involved in peripheral replication. They state that the impact of such variants on human health cannot be estimated by epidemiological surveys relying upon current diagnostic criteria for vCJD and so they suggest that these criteria should be adjusted accordingly. They conclude that the range of incomplete syndromes that they observed between healthy carriers and typical vCJD indicates that multiple forms of prion variants can coexist and may emerge in different forms depending upon the conditions under which transmission occurred. Overall this is a carefully written manuscript with suitable statistics and well-presented data in the main body of the text. The main conclusion from the manuscript that "multiple forms of prion variants can coexist and may emerge in different forms depending upon the conditions under which transmission occurred" is not a new phenomenon. Other groups have stated this in the past and prion biologists have been aware of this for a while. Having said that, there are some new and very interesting findings in this current manuscript.

We agree with the referee that the coexistence of different prion variants/strains is a well-known notion from prion biologists. However, in the previous studies illustrating this phenomenon, the emerging prion diseases always harbor the classical specific hallmarks of TSEs as spongiform changes and accumulation of pathological PrP in brain. Our study conversely brings two major novelties: the restriction of lesions to spinal cord without cerebral involvement on one hand, and the occurrence of prion diseases devoid of the prion-specific hallmarks on the other hand. In other words, we describe here new pathological entities that escape the diagnosis of prion diseases on the basis of the current criteria, contrarily to the aforementioned variants.

There is however a need to shorten the amount of supplementary data that supports this paper. It is not necessary or useful to have 35 pages of supplementary data and less pages in the actual document to be published. I do feel there is a need to shorten and condense the amount of supplementary information so that the manuscript is more self-contained and less cumbersome.

We agree with the reviewer #3 that we provided many information in this manuscript. Another reviewer formulated similar comment. We lightened the supplementary information by transferring three supplementary figures, one supplementary table and one supplementary note in the main text, and merged three supplementary figures in one.

Our results are based on primary exposure of primate and mice to brain and blood samples, and secondary transmission to mice. We previously submitted the primate part of those data as single, but it was judged as insufficient, and we believe that all those different independent experiments are like Hercules' columns that strengthen each other to be fully demonstrative. The only set of data that eventually might be removed are the alternative etiologies that we explored. However, these explorations were systematically required by our scientific colleagues when we presented our observations on primates, to eliminate obvious non-prion cause to this myelopathic syndrome. We made the choice to keep these information as a supplementary note, and also the detailed experiments on primate transmission and secondary transmission on mice, that are information dedicated to specialists.

Major point: The manuscript is interesting and has thrown up some important information regarding

incomplete syndromes that require further analysis and study in order to clarify some of the points raised by the authors. I agree there are obvious consequences for public health and there is a need for this type of work to highlight the issues.

I appreciate the vast amount of work involved and the fact that prion studies can take up a huge amount of time (especially primate studies). Some of the evidence that has been given is inconclusive and so requires further analysis. Much of the content comes over as not being complete or needing a bit more experimentation or analysis in order to generate conclusive information.

We agree with the reviewer that this is a long story that is not ended. All this work corresponds to ten years of work on primates and mice, and a point to date needs to be made at a moment to bring this important information to the medical and scientific community. Further studies are ongoing on the different samples we have collected, notably to develop techniques allowing the evidence of abnormal PrP^{res} deposition in PrP^{res} negative phenotypes but this will be the topic of other publications. Concerning mice, even if 80% of the animals were subject to both biochemical and histological analyses, some of them (20% of clinically-affected animals after first passage) could unfortunately not be sampled for histology but only for biochemistry: complete diagnosis cannot be performed. However, it should be reminded that in human the situation is much more worse, as the autopsies are very rare.

Minor point: Numbering lines from 1 on every page is actually not helpful.

I am slightly disturbed by the word “.....considered to be...” such as in lines 21 and 25 (page 6). This does not demonstrate a definite result and so it needs further study.

Every now and then there are phrases that make me slightly nervous of the robustness of the data shown.

I have an issue with what the authors have said on page 7 lines 24 ad 25. To say an animal “might exhibit....” is hardly scientific and needs much better explanation or clarification.

All the mice included in this study were tested for the presence of PrP^{res} accumulation in their brain through biochemical approaches, which is currently the main criterion for diagnosis of prion diseases. For a vast majority of them, half of the brain was also sampled to confirm the presence of spongiform changes but some of them could not be sampled for histology.

- Seven of those animals not sampled for histology had clinical signs and were found PrP^{res} positive. They should then be classified as vCJD. However, according to our observations, these animals may have developed either a vCJD or a bulbospinal phenotype, but the absence of samples for histology prevents us a formal conclusion. The scientific rigor imposes to count them as vCJD. We used the wording “considered to be” since it seemed to us to be the appropriate wording, but we accept to change it. We thus modified the sentence (page 6 line 21) “*Seven other animals, that showed accumulation of PrP^{res} but were not sampled for histology, were also considered to have developed vCJD (total number of vCJD cases was thus considered to be 36)*” in “*Seven other animals, that showed accumulation of PrP^{res} but were not sampled for histology, were classified as vCJD (total number of vCJD cases was thus 36)*”.
- 13 of those animals not sampled for histology also exhibited clinical signs but were devoid of PrP^{res} accumulation. In the absence of histological analysis, they cannot be definitely classified with a NL, B, C or S phenotype. We might have removed them from the study, but this would have modified the total count, and most of all they transmitted prion diseases upon serial passage as

illustrated in the panel a of the supplementary figure 4. We thus preferred to maintain them in the total count and classified them as “non vCJD” animals. We modified the sentence (page 7 line 24) *“The 13 remaining PrPres- animals were not sampled for histology and were thus classified as “non vCJD”, but they might exhibit an S, C, B or NL phenotype.”* In *“The 13 remaining PrP^{Res} negative animals could not be sampled for histology and thus cannot be specifically subclassified within the S, C, B or NL phenotypes. They were thus classified as “non-vCJD”*

The sentence (page 6 line 25) *“Those disorders could be considered to be truncated vCJD phenotypes as they presented similar lesions but not the complete spectrum”* was modified as *“Those disorders appeared as truncated vCJD phenotypes as they presented similar lesions but not the complete spectrum”*

I feel the paper needs condensing by removing some of the supplementary data and text and it needs some clarification in places where animals were not tested by one or more protocols and were then “considered to be....”. The samples either need to be tested fully or the text needs adjusting to make things clearer, less ambiguous and more scientific.

This paragraph summarizes the different points underlined by the referee, we answered to them above.

REVIEWERS' COMMENTS:

Reviewer #1 (Remarks to the Author):

With this careful revision the authors have effectively addressed my previous concerns and improved the manuscript.

Reviewer #2 (Remarks to the Author):

The authors have appropriately answered my concerns. I believe this article provides important information.

Reviewer #3 (Remarks to the Author):

The authors have made fair and sensible responses to the reviewer's comments and have attempted to address the points raised. There has been a reduction in the total number of figures displayed and modifications to the huge amount of Supplementary material that was presented initially. This has made a significant improvement to the overall presentation of the data.

I fully appreciate the need to appeal to the 'dedicated specialist' and I think that is still the case but the manuscript is now much more concise and better presented for all readers (specialist or otherwise).

The authors have clearly explained their reasoning and I fully understand that this manuscript is the culmination of many years work. The document is now in a format that allows for publication.

I am happy that all the reviewer's points have been addressed or at least commented on in a constructive manner and I think the final version is now much more manageable on a scientific level.

I look forward to reading their future publications based on their current, on-going experiments.